# Self-Powered Flexible Sour Sensor for Detecting Ascorbic Acid Concentration Based on Triboelectrification/Enzymatic-Reaction Coupling Effect

**DOI:** 10.3390/s21020373

**Published:** 2021-01-07

**Authors:** Tianming Zhao, Qi Wang, An Du

**Affiliations:** College of Sciences, Northeastern University, Shenyang 110819, China; zhaotm@stumail.neu.edu.cn (T.Z.); wangqi@mail.neu.edu.cn (Q.W.)

**Keywords:** ascorbic acid detection, self-powered, sour sensor, triboelectrification/enzymatic-reaction coupling effect

## Abstract

Artificial sensory substitution systems can mimic human sensory organs through replacing the sensing process of a defective sensory receptor and transmitting the sensing signal into the nervous system. Here, we report a self-powered flexible gustation sour sensor for detecting ascorbic acid concentration. The material system comprises of Na_2_C_2_O_4_-Ppy with AAO modification, PDMS and Cu wire mesh. The working mechanism is contributed to the triboelectrification/enzymatic-reaction coupling effect, and the device can collect weak energy from body movements and directly output triboelectric current without any external power-units. The triboelectric output is affected by AA concentration, and the response is up to 34.82% against 15.625 mM/L of AA solution. Furthermore, a practical application in detecting ascorbic acid concentration of different drinks has been demonstrated. This work can encourage the development of wearable flexible electronics and this self-powered sour sensor has the potential that can be acted as a kind of gustatory receptors to build electronic tongues.

## 1. Introduction

The internet of things (IoT) is a huge network integrated with sensors in order to interface with our daily lives through data exchanging [1,2,3,4,5]. Furthermore, in recent years, the establishment of body-electric interfaces has attracted notable attention via multifarious electronic skins for capturing indexed in big data analytics [6,7,8,9,10]. Some devices can detect toxic agents in the environment and define the quality of the food, and some devices can reflect the human body movement state and physiological information [11,12,13,14,15,16,17,18,19,20,21,22,23,24,25,26]. For example, sensors for gustation recognition are immersed in the measured solution, and the changes of the electrical signal can recognize the flavor [27,28,29,30,31]. These traditional sensors are usually based on potentiometer, voltammetry and impedimetric titration, which have a wide range of detections [32,33,34,35,36,37]. However, the power-supply units integrated with the sensing systems limit the development of innovative portable gustation devices. The bulky volume of the sensing systems reduces the comfort level and low capacity and frequent charge/discharge process of the power-supply units increase the risk for safe problems, which are still bottlenecks to be overcome [38,39]. Thus, a new kind of gustation electronic tongue is given an opportunity to build self-powered system in day-to-day application.

Recently, the wearable piezoelectric/triboelectric nanogenerators as a self-powered system can actively output electrical signal though harvesting tiny mechanical energy from the human body and the electrical signal can reflect the human body movement state or physiological information at the same time [40,41,42,43]. Furthermore, through modifying enzymes, the piezoelectric/triboelectric process can be affected by enzymatic reactions. In our previous work, self-powered electronic skins by modifying glucose oxidase for sensing glucose in sweat has been obtained [44,45,46]. Basing on triboelectrification/enzymatic-reaction coupling effect, the triboelectric outputting current signal can act as not only electric energy but also biosignal which is depended on the glucose concentration. Thus, by redesigning the structure of the device and material system, a new self-powered gustation electronic tongue for sensing sour taste can be achieved.

In this paper, a self-powered flexible sour sensor for mimicking taste buds has been put forward through a simple way. This self-powered sour sensor is fabricated from Na_2_C_2_O_4_ doped polypyrrole/polydimethylsiloxane (Na_2_C_2_O_4_-Ppy/PDMS) nanostructures. PDMS and Na_2_C_2_O_4_-Ppy have reported that both materials have excellent biocompatibility. In addition, Na_2_C_2_O_4_-Ppy are synthesized through electrochemical polymerization process. Ascorbate acid oxidase (AAO) is modified on the surface of Ppy layer. Basing on triboelectrification/enzymatic-reaction coupling effect, this sour sensor can convert weak energy into the current signal and this signal is markedly dependent on the concentration of ascorbic acid. Furthermore, the whole process does not involve any extra power source. At last, a practical application for sensing ascorbic acid concentration five samples has been demonstrated. Our work can encourage the development of wearable flexible electronics and develop a new direction for building electronic tongues.

## 2. Materials and Methods

### 2.1. Materials

Polydimethylsiloxane (PDMS) and copper wire mesh were purchased from Taobao. Pyrrole was bought from Sinopharm Chemical Reagent Co., Ltd. (Beijing, China). Sodium oxalate (Na_2_C_2_O_4_), sodium persulfate, phosphate buffer saline (PBS), glucose, uric acid (UA), urea and ascorbic acid (AA) were purchased from Shanghai Aladdin Biochemical Technology Co. Ltd. (Shanghai, China). Ascorbate acid oxidase (AAO) was provided by Shanghai Lanji Technology Development Co., Ltd. (Shanghai, China).

### 2.2. Device Fabrications

Copper wire mesh was washed by deionized water and alcohol several times to remove the impurities and dried in N_2_ flow at 60 °C. PDMS was mixed with a mass ratio of 10: 1 by elastomer and cross-linker under ultrasonic bath (40 kHz, 80 W) for 30 min at room temperature (~20 °C). Then the mixture was placed in a vacuum oven to obtain the air-free PDMS. The air-free PDMS colloid was poured into a box and put in a vacuum oven at 90 °C for 8 min. Next, the precleaned copper wire mesh was cut into pieces (2 cm × 5 cm) and put on the PDMS before being completely solidified. Finally, the copper-wire-mesh/PDMS film was solidified in a vacuum oven at 90 °C for 2 h. It was worth highlighting that the first-curing-process must be conducted and in this case, the copper wire could be immobilized on the surface of the PDMS rather than embedding in the PDMS layer. After solidification process, the film was immersed into the sodium persulfate aqueous solution (0.1 m/L) for 30 s (wet-etching process). The transient wet-etching process would remove less Cu and Cu mesh that was still on the surface of PDMS. Furthermore, this process aimed to form growth space for electrochemical polymerization of Na_2_C_2_O_4_ doped polypyrrole (Na_2_C_2_O_4_-Ppy). The Na_2_C_2_O_4_-Ppy on the surface of copper wire mesh was synthesized via electrochemical polymerization. Cu mesh of the device was used as working electrode to deposit Na_2_C_2_O_4_-Ppy, Ag/AgCl was used as the reference electrode and Pt wire was used as the counter electrode, respectively. Cyclic voltammetry mode was conducted from 1.2 V to −1.2 V for 400 s and the polymerization solution contained 0.1 M/L pyrrole monomer and 0.2 M/L Na_2_C_2_O_4_. Then, the Cu film (~200 nm) was deposited by electron beam evaporation equipment. Finally, the surface was washed by deionized water for several times and dried overnight.

Ten milligrams AAO was dissolved in 10 mL PBS. Then 0.5 mL AAO solution was dropped on the device for four times and an incubation procedure was conducted in a fume hood for 8 h. The prepared devices were stored at 4 °C.

### 2.3. Characterization and Measurement

The microstructure of the device was examined by Scanning electron microscopy (SEM, Hitachi S4800). The electron beam evaporation equipment (DZS500, Pengcheng Vacuum Technique, Inc., Xuzhou, China) was used to deposit Cu back electrode. The electrochemical workstation (CHI627D, CH Instruments, Inc., Austin, TX, USA) was used to synthesize the Na_2_C_2_O_4_-Ppy. The measurement system contained a controller and an actuator, which can set driving force, moving speed, moving distance and cycling time. Appendix A showed the photograph of the measurement system. The data was collected by Stanford SR 560 (a low-noise preamplifier). The temperature was conducted at ~20°C and the relative humidity was kept at 40%. Each kind of measurement was replicated 10 times.

## 3. Results

### 3.1. Experimental Design

As shown in Figure 1a, approximately 10,000 taste receptors named as taste buds grow on the surface of the tongue. These taste buds can be stimulated during chewing course and transport bioelectric signal to the specific encephalic region through afferent fibers, telling what the flavors are (Figure 1b). The design of self-powered flexible sour sensor can collect weak energy from body movements and actively output current signal. By modifying AAO, the enzymatic reaction can control the triboelectrification process and the output triboelectric current is depended on AA concentration. The friction materials are PDMS and Ppy. PDMS and Ppy are reported by their excellent biocompatible and considerable gap of electronegativity. PDMS can easily capture electrons from Ppy and leave the equal numbers charges on the surface of Ppy layer [47,48,49]. Thus, this self-powered flexible sour sensor overcomes the bottleneck of the power source. Furthermore, the fabrication of the device is shown in Figure 1c. The process of fabrication contains solidification, wet-etching, electrochemical polymerization, depositing electrodes and modifying enzyme. More details can be seen in Section 2. It is worth mentioning that the solidification time must be strictly obeyed. Na_2_C_2_O_4_-Ppy can be only deposited on half-submerged copper wire mesh. A large area device can be obtained in this simple way, and the large area device can be cut into suitable pieces for measurement. Figure 1d shows the measurement system, containing a controller and an actuator, which can set driving force, moving speed, moving distance and cycling time. The device is nailed to the actuator. When the actuator is forward (Figure 1di), the device will be deformed and when the actuator is backward, the device will be restored to the original state (Figure 1dii). The triboelectric current output when the device is under deformation and the data can be collected by SR 560. The different concentration of AA, uric acid (UA), glucose and urea solutions are dropped on the surface of the devices for the measurements. Appendix A shows the photograph of the measurement system and Appendix A shows the details of the device. The working frequencies and angles can be calculated according to the driving force, moving speed, moving distance and cycling time.

### 3.2. Characterization of the Self-Powered Sour Sensor

Figure 2 demonstrates the images of self-powered sour sensor. Figure 2a shows the SEM image of copper-wire-mesh/PDMS film. It can be manifestly seen the copper wire mesh is half immersed in the PDMS layer, which should be firmly fixed in PDMS. Figure 2b shows the SEM image of copper-wire-mesh/PDMS film after wet-etching. After the wet-etching process, there is a gap between PDMS and copper wire mesh, providing growth space of the Na_2_C_2_O_4_-Ppy. Furthermore, this gap provides sufficient interval between the Na_2_C_2_O_4_-Ppy and PDMS for friction. Figure 2c shows the SEM image rub with of the device after electrochemical deposition. A rough surface of the Ppy film can effectively PDMS layer, increasing the triboelectrification process. Figure 2d shows an optical image of the device. The device is so adaptable to fit human skin.

## 4. Discussion

### 4.1. Sour Sensing Performances

Self-powered flexible sour sensor for AA detection can actively output triboelectric current signal which is dependent on AA concentration and the AA biosensing performance has been presented in Figure 3. The self-powered sour sensor is connected to the external circuit for testing the AA biosensing performance against the AA concentration from 0.005 to 15.625 mM/L. All the experimental measurements are carried out at room temperature. The deformation on the device is conducted by a stepping motor, of which the movement can be monitored by programming. The bending angles and frequencies are programmed to 15° and 1 Hz, respectively. As shown in Figure 3a, the output triboelectric current is markedly dependent on the AA concentration, and the triboelectric current signal decreases with the increasing AA concentration. As the concentration of AA is 0.005, 0.025, 0.125, 0.625, 3.23 and 15.625 mM/L, the output triboelectric current is 6.67 ± 0.20, 5.56 ± 0.14, 5.13 ± 0.08, 4.85 ± 0.16, 4.62 ± 0.08 and 4.35 ± 0.10 nA, respectively. Figure 3b shows enlarged views of output triboelectric current at the AA concentration of 0.0025 and 3.125 mM/L, respectively. It can be seen from the stability of the output triboelectric current at different AA concentration. A control experiment is designed to verify the triboelectrification/enzymatic-reaction coupling effect of the device, as shown in Figure 3c,d. Figure 3c shows the relationship between AA concentration and output triboelectric current of the device with AAO modification, and the red line (from 0.025 to 15.625 mM/L) is a linear fit. The linear fitting of Equation is as follows:(1)y = 4.81−0.40 ×lg(x),
where y represents the triboelectric current (nA) and x represents the AA concentration (mM/L). Furthermore, the linearity is up to 0.985. The upper limit of detection is 15.625 mM/L and the lower limit of detection is 0.0025 mM/L. Figure 3d shows the relationship between AA concentration and output triboelectric current of the device without modifying AAO. It can be manifestly observed that the device without AAO modification does not have the AA sensing performance. The totally distinct behaviors of the two devices indicate that enzymatic reaction can control the triboelectrification process. The output triboelectric current can be regarded as a function of AA concentration, as shown in Figure 3e. The response of the device can be calculated from the following equation:(2)$$R\%=\frac{\left|I_{0}-I_{t}\right|}{I_{0}}\times 100\%,$$
where *I*_0_ and *I_t_* represent the output triboelectric current in 0.005 mM/L and others AA concentration solution, respectively. For the device with AAO modification (Group 1), as the concentration of AA is 0.005, 0.025, 0.125, 0.625, 3.125 and 15.625 mM/L, the corresponding response is 16.74%, 23.21%, 27.31%, 30.71% and 34.82%, respectively. Furthermore, for the device without AAO modification (Group 2), as the concentration of AA is 0.005, 0.025, 0.125, 0.625, 3.23 and 15.625 mM/L, the corresponding response is 3.02%, 4.80%, 4.30%, 0.04% and 1.19%, respectively. As shown in Figure 3f, only adding water on the surface of the device with AAO modification, the triboelectric is almost a constant. In addition, the response is 0, 2.03%, 0.03%, 0.3%, 0.03% and 0.8%, respectively. Figure 3g shows the response of the device with AAO modification during 8 days against AA solutions (15.625 mM/L). The response is 34.82%, 35.42%, 30.41%, 22.14% and 10.98% in five days and after replenishing AAO, the response restore to 36.41%, 34.14% and 32.64%. The device is effective for 3–5 days due to the activity of AAO. However, this problem can be solved by replenishing AAO regularly.

Figure 4 shows the influence of the bending angles on the AA biosensing performance of the self-powered flexible sour sensor. The concentration of AA is conducted in 0.005 and 3.125 mM/L, respectively. Figure 4a shows the output triboelectric current of self-powered flexible gustation sour sensor under different bending angles. As the bending angles are 15°, 30°, 45° and 60°, the output triboelectric current against 0.005 mM/L of AA solution (*I*_0_) is 6.96 ± 0.47, 5.34 ± 0.28, 4.33 ± 0.17 and 3.48 ± 0.16 nA, and the output triboelectric current against 3.125 mM/L of AA solution (*I_t_*) is 5.83 ± 0.32, 4.18 ± 0.45, 3.70 ± 0.28 and 2.50 ± 0.28 nA, respectively (Figure 4b). The relationship between response and angle can be seen in Figure 4b. As the bending angles are 15°, 30°, 45° and 60°, the response of the device against 3.125 mM/L of AA solution is 16.27%, 27.74%, 14.5% and 28.15%, respectively. Though the output triboelectric current decreases with the increasing bending angles, the response is almost the same. These results show that self-powered flexible sour sensor have excellent flexibility and stability for its practical applications.

### 4.2. Sensing Mechanism

To further confirm the sensing mechanism for AA, some common compounds in body have been tested. As shown in Figure 5a, increasing the concentration of the uric acid solution cannot lower the output triboelectric current. Furthermore, the similar results can be observed in Figure 5b,c. The device modified with AAO cannot detect urea and glucose. These results imply that the lowered output triboelectric current of the devices is due to the AA concentration. Only increasing the concentration of the AA solution can decrease the output triboelectric current. Furthermore, the increasing concentration of uric acid, glucose and urea cannot influence the output triboelectric current of the self-powered flexible sour sensor.

The working mechanism of self-powered flexible sour sensor is shown in Figure 6. Figure 6a,b show self-powered flexible sour sensor and one sensing unit of the device. Blue part, green part and yellow part represent PDMS, Ppy and Cu, respectively. Furthermore, the red torus knot represents AAO. After incubation procedure, AAO is immobilized onto the surface of Ppy. The triboelectrification process between PDMS and Ppy is shown in Figure 6c [36,50]. In the initial state, the PDMS layer and Ppy layer start to rub. Due to the triboelectric effect, the charges accumulate on the contact area. Because PDMS is more electronegative than Ppy, negative charges accumulate on the PDMS layer and the positive charges accumulate on the Ppy layer. Then, the PDMS and Ppy are separated from each other due to the deformation, and the charges are contained in PDMS layer and Ppy layer, respectively. With the increasing separated distance, the positive charges migrate from the Ppy layer to the Cu back electrode via the external circuit under the electrostatic field. Furthermore, when the separated distance increase to the max, the charges stop migrating from Ppy layer to the Cu back electrode. When the PDMS layer and Ppy layer come into contact due to the deformation, the charges move in reverse (from Cu back electrode to Ppy layer). During the electricity generation process, the output can be detected by Stanford SR 560 via the external circuit. The theoretical real-time electricity generation of the triboelectrification effect can be described by the equations as follows [51,52,53]:(3)$$Q_{SC}=\frac{\sigma_{i}Sg\left(t\right)}{d+x},$$
where *Q_SC_* represents the triboelectric charge; *σ_i_* represents the triboelectric charge density of Ppy; *S* represents the contact area; *d* represents the separation distance; *g*(*t*) represents the separation distance between the copper wire mesh and Ppy (a function of time); *x* represents the maximum of the separation distance; and *I_SC_* represents the theoretical outputting current. 

The enzymatic reaction can control the triboelectrification process and Figure 6d shows the production of the enzymatic reaction of AA and AAO [54]. When the enzymatic reaction is carried out, Ppy is exposed to H^+^ ions. Due to the deprotonation behavior of Ppy, the surface chemical state transforms into Ppy_ox_. During the triboelectrification process, the gap of electronegative between PDMS and Ppy reduces and fewer charges can be driven through the external circuit, which reduces the output triboelectric current. Furthermore, with the AA concentration increasing, the more H^+^ ions release from the enzymatic reaction, resulting in the lower output triboelectric current. Thus, the output triboelectric current can be regarded as both power and biosensing signal and the device does not need any external power source.

### 4.3. Applications

Figure 7 shows a practical application of this self-powered flexible sour sensor for detecting ascorbic acid against mineral water (Figure 7a), apple juice (Figure 7b), compound juice (Figure 7c), orange juice (Figure 7d) and vitamin drink (Figure 7e), respectively. These drinks can be bought from shops and the concentration of ascorbic acid can be found in ***Nutrition Facts***. Furthermore, the AA concentrations of these drinks have been shown in Appendix A. The AA concentration of apple juice is 0.426 mM/L (Appendix A), the AA concentration of compound juice is 1.275 mM/L (Appendix A), the AA concentration of apple juice is 0.426 mM/L (Appendix A) and the AA concentration of vitamin drink is 1.136 mM/L (Appendix A), respectively. The solutions are dropped on the surface of the self-powered sour sensor, and the triboelectric actively output under the deformation. Figure 7a–e shows the triboelectric current against mineral water (Figure 7a), apple juice (Figure 7a), compound juice (Figure 7c), orange juice (Figure 7d) and vitamin drink (Figure 7e), respectively. The triboelectric current is 4.98 ± 0.22, 4.71 ± 0.48, 4.94 ± 0.26 and 4.79 ± 0.12 nA against mineral water, apple juice, compound juice, orange juice and vitamin drink, respectively. As shown in Figure 7f, the concentrations of these drinks are measured and the red line is actual value and green line is experimental value. The concentrations of these drinks are calculated using Equation (1). The experimental concentrations are 0.362, 1.775, 0.484 and 1.142 mM/L against apple juice, compound juice, orange juice and vitamin drink, respectively. It can be seen that the experimental value and actual value of these drinks are similar. It should be noted that all the solutions are filtered with filter papers before the measurements. Otherwise, the juice residues would lower the triboelectric output and influence the sensing process. These results suggest that this self-powered flexible sour sensor can sense AA concentrations and, in the future, it can be acted as gustatory receptors for building electronic tongues.

## 5. Conclusions

In summary, self-powered flexible sour sensor for detecting ascorbic acid concentration has been fabricated from Ppy/PDMS structure. The material system consists of Na_2_C_2_O_4_-Ppy with AAO modification, PDMS and Cu wire mesh. The sour senor can convert weak energy from body movement into current signal and the biosignal is markedly affected by the concentration of ascorbic acid. Basing on the triboelectrification/enzymatic-reaction coupling effect, the response of the device is up to 34.82% against 15.625 mM/L AA solution. Furthermore, the whole process does not involve any extra power source. A practical application has been demonstrated for detecting ascorbic acid concentration of different drinks. This study reports an important advancement in low-cost self-powered nanosystems.

## Figures and Tables

**Figure 1 sensors-21-00373-f001:**
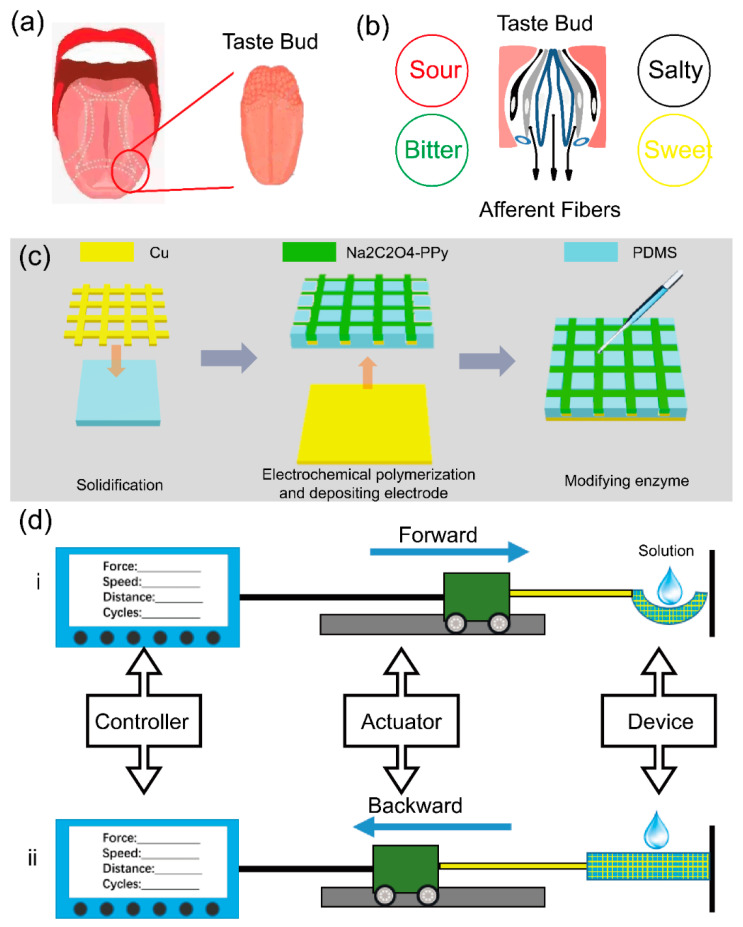
(**a**) Taste receptor: taste bud. (**b**) Gustation recognition. (**c**) The fabrication of the device. (**d**) The measurement system.

**Figure 2 sensors-21-00373-f002:**
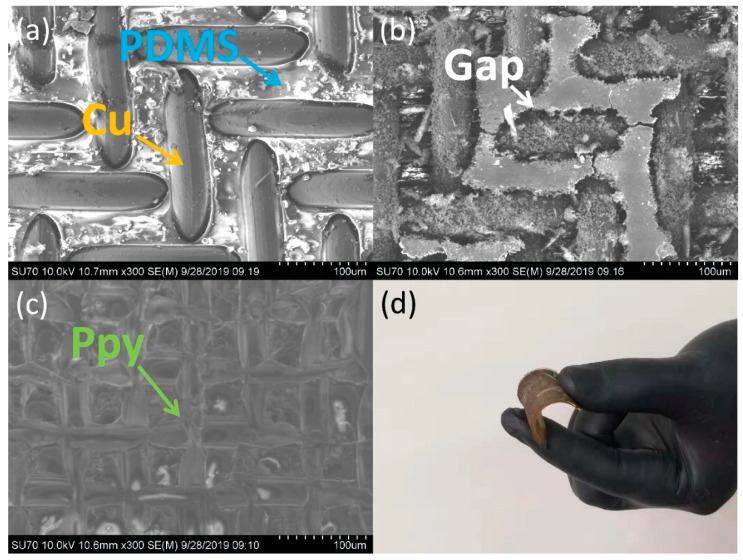
The characterization of self-powered sour sensor. (**a**) SEM image of copper-wire-mesh/PDMS film. (**b**) SEM image of copper-wire-mesh/PDMS film after wet-etching. (**c**) SEM image of the device after electrochemical deposition. (**d**) The optical image of the self-powered sour sensor.

**Figure 3 sensors-21-00373-f003:**
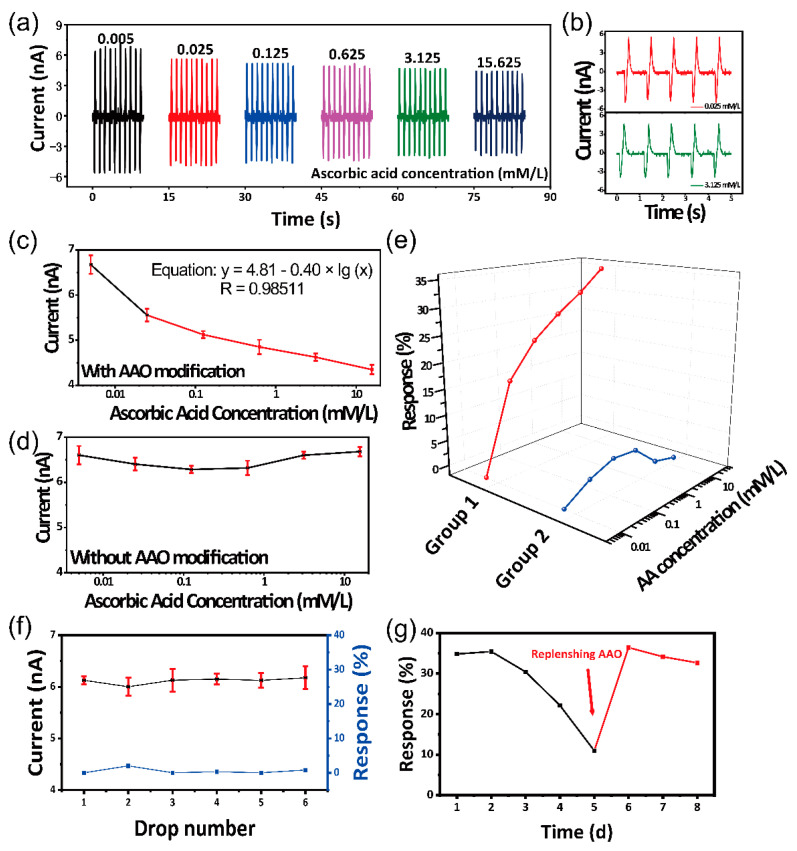
The AA biosensing performance of the self-powered sour sensor. (**a**) The output triboelectric current of the device against AA with the concentration from 0.005 to 15.625 mM/L. (**b**) The enlarged views of output triboelectric current at the AA concentration of 0.0025 and 3.125 mM/L, respectively. (**c**) The output triboelectric current of the device with AAO modification against AA with the concentration from 0.005 to 15.625 mM/L, respectively. Red line is the limit of detection. (**d**) The output triboelectric current of the device without AAO modification against AA with the concentration from 0.005 to 15.625 mM/L, respectively. (**e**) Response of two control experiments. (**f**) The triboelectric current and response of the sour sensor by adding deionized water. (**g**) The response of the sour sensor during 8 days.

**Figure 4 sensors-21-00373-f004:**
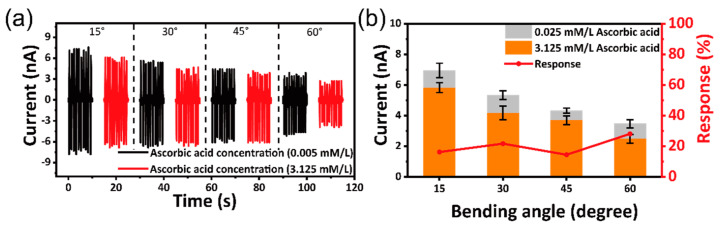
(**a**) Output triboelectric current of self-powered flexible sour sensor under different bending angles. (**b**) The relationship between bending angles and response.

**Figure 5 sensors-21-00373-f005:**
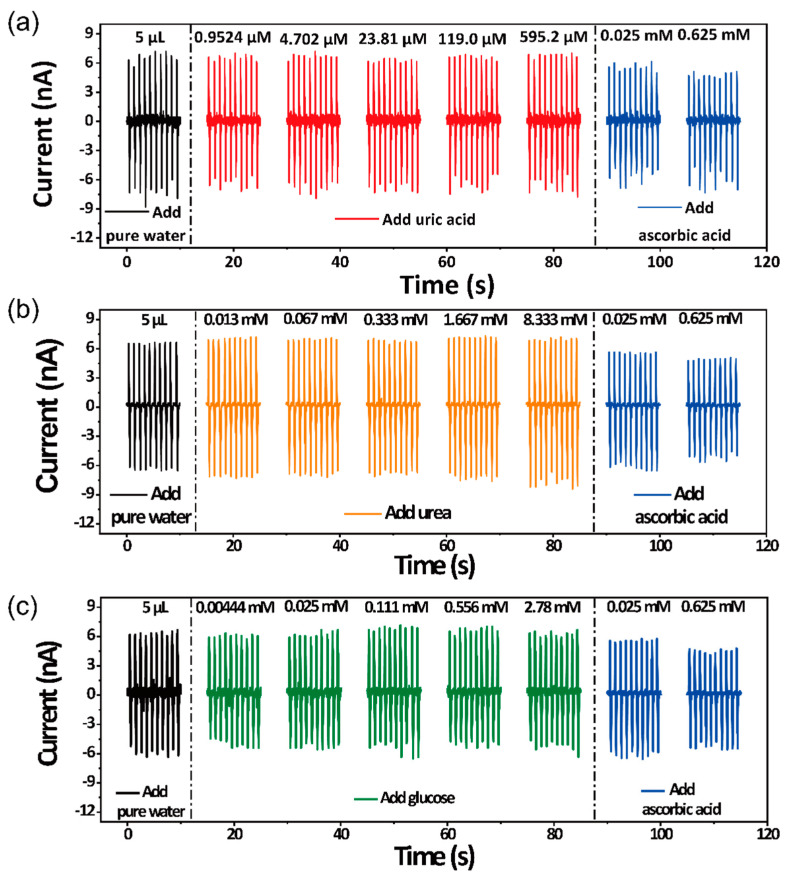
Control experiments. (**a**) Influence of different uric acid concentrations. (**b**) Influence of different glucose concentrations. (**c**) Influence of different urea concentrations.

**Figure 6 sensors-21-00373-f006:**
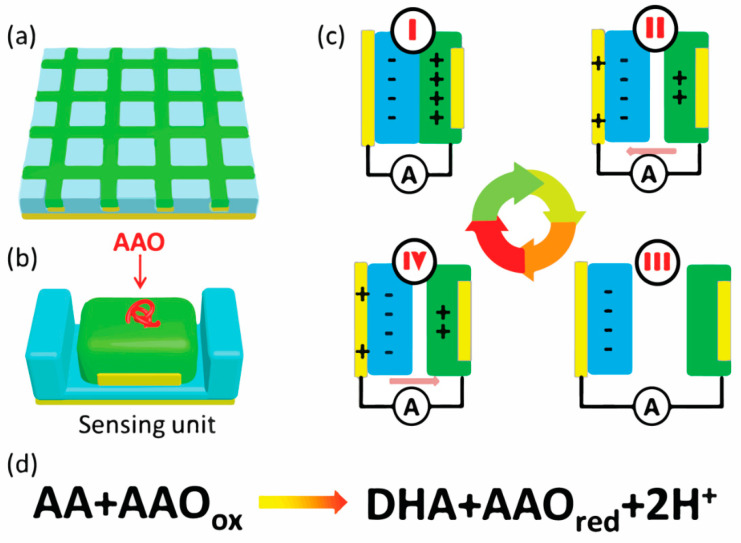
The working mechanism of the self-powered flexible sour sensor. (**a**) The self-powered flexible sour sensor. (**b**) One sensing unit. (**c**) The electricity generation process. (**d**) The enzymatic reaction: AA and AAO.

**Figure 7 sensors-21-00373-f007:**
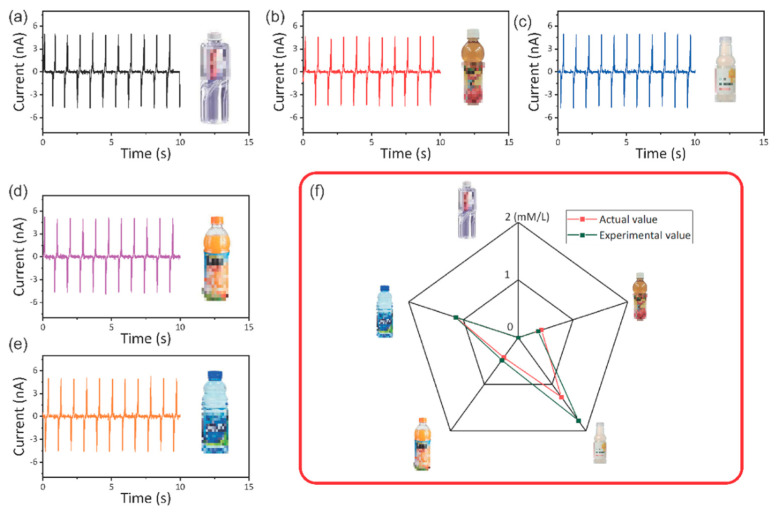
The practical application in detecting ascorbic acid in different drinks. (**a**–**e**) The output triboelectric current against mineral water, apple juice, compound juice, orange juice, and vitamin drink, respectively. (**f**) Corresponding measured response and AA concentration in these drinks.

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
