# Peer review of "Self-Powered Flexible Sour Sensor for Detecting Ascorbic Acid Concentration Based on Triboelectrification/Enzymatic-Reaction Coupling Effect"

_sensors, 2021, doi:10.3390/s21020373_

Round 1

Reviewer 1 Report

In this paper, a m electronic tongue dedicated to the detection of ascorbic acid based on triboelectrification/enzymatic reaction coupling effect is reported.

The objectives and results are clearly interesting, however, the paper has several flaws that must be corrected before the paper is published.

The main concern is that the preparation of the device is not well explained and many aspects remain obscure.

  1. Regarding PPy, the polymerization of Ppy is difficult to understand. Many more details about polymerization and doping of the Ppy with sodium oxalate must be given.
    • Have authors used a previously electropolymerized PPy that is deposited on the top of the surface?. Or, the electropolymerization has been carried out in situ?. In any case, details of the electropolymerization must be given:  indicate the electrochemical method used  (CA, CP, CV..); Voltage or intensity applied and duration,;  concentration of the reactants and doping agent. Nature of the reference, counter electrode and working electrode must be given.
    • If the electropolymerization is carried out in situ, even more explanations are needed:  Taking into account that PDMS is a non conducting material, I assume that these copper wires act as the working electrode. However, this is not the case because in Line 103 authors claim that  “After the etching there is a gap between PDMS and copper wire mesh, providing growth space of the Ppy”. This needs to be explained.  It seems that Ppy growth in the space between PDMS and copper wire mesh. Why? Why Ppy does not growth onto copper?. How is the device connected to the potentiostat to produce the electrodeposition?.
    • Explain in detail how the enzyme has been deposited and immobilized onto the device surface (concentration, volume, method used for deposition, drying, etc..)
  2. Line 48, line 66, line 80. It is written “gustation” “flavors”, etc. There is a general agreement in the scientific community that processes described here have nothing to do with human feelings of taste. As they are just chemical reactions the terms gustation, flavor and other similar terms should not be used (the classical examples are MgCl2 and quinine, both having bitter taste and producing completely different signals)
  3. Line 78 In Figure 1 a “huge amount of taste receptors”. In science, the terms big, small large, huge make no sense. Please quantify the number of receptors
  4. Line 85: Please, explain the meaning of “considerable gap of electronegativity”.
  5. Figure 3: The response in the absence of AA must be shown. In the proposed mechanism, described on page 8, H+ play an important role. However, AA is also an organic acid. That can affect the response of Ppy
  6. Data of repeatablity and reproducibility must be given

Other issues.

They are several of spelling errors that must be corrected. Some examples are:

  • Figure 1 “Electrochimical” must be changed to “Electrochemical”
  • Line 53: Materilas must be changed to materials

The English must be corrected. The meaning of certain sentences is hard to understand 

Figures are not well placed

Reviewer 2 Report

The paper aimed to develop a self-powered flexible gustation electronic tongue for mimicking taste buds has been which, does not involve any extra power source and can act as wearable flexible electronics and artificial neural networks.

First of all, authors should clearly define the basic components of an electronic tongue instrument and what instruments/devices qualify to be called an electronic tongue. Many times of electronic tongues exist e.g potentiometric, voltammetric, impedemetric etc. How would you define what you developed?

The current descriptions in the materials and methods section  is too simplified and makes the paper rather technical. Thus, the materials and methods section of the paper needs major improvement. Also, the full meaning of some the materials used in the construction of this gustation instrument should be written when they appear for the first time. E.g polypyrrole (PPy)/polydimethylsiloxane (PDMS). Lines 92-107 should be at the methods sections as it explains how the device was constructed not the result it gives.

It was written in the paper that the newly constructed device was used to measure AA and AAO but it is not clearly written how the measurement was done. Was the instrument put on the skin during measurement? Which part? How was the sample tasted with the instrument? How many times were the samples tasted? Were the samples in solution? If yes, how were the solutions prepared, if no what was their physical state and how was the measurement done? What measures were taken to overcome environmental influence (temperature, humidity etc) during the environment? All the steps of measurement should be explained. If possible pictures, flowcharts or figures can be included for easy understanding.

Authors should be consistent with the use of expressions such as “self-powered flexible gustation electronic skin” (line 161) and “Self-powered flexible gustation electronic tongue” (line 162). I noticed many parts of the results and discussion ought to actually be in the materials and methods section. The paper needs restructuring. In addition, it is not in the SENSORS format (introduction, materials and methods, results, discussion). The results should be separated from the discussion. Figure 7 apparently shows practical applications of the developed device using juice samples. How were these measurements performed? The whole paper should be thoroughly checked for English corrections.

Reviewer 3 Report

The manuscript by Tianming Zhao et al. titled “Self-powered flexible electronic tongue for detecting ascorbic acid concentration based on triboelectrification/enzymatic-reaction coupling effect” reports an enzymatic electrochemical sensor for detection of ascorbic acid powered by the triboelectric effect.

The idea of this work, i. e. development of self-powered enzymatic biosensor, is very interesting. However, some important information is missing in the presentation of the results and experimental and some statements are questionable. The manuscript needs to be rewritten prior to be considered for the publication in the Sensors.

Comments.

  1. The authors call developed sensor “electronic tongue” and state that sensor mimics gustatory receptors. This is not correct. Biological gustatory receptors have nothing in common with developed sensors either in the composition or in the sensitivity. Developed sensor selectively detects concentration of one compound, ascorbic acid, it does not detect taste. Electronic tongues as well as gustation comprise several sensors or receptors with varying sensitivity and specificity. Thus, one single sensor can not be called electronic tongue. Eventually, sensors based on the triboelectric effects can be used for construction of the electronic tongue but it is not what was done in this work.
  2. The authors write in the abstract “This work can encourage the development of wearable flexible electronics and artificial neural networks”. Artificial Neural networks are unrelated to this work.
  3. 1, lines 29-30. Ref. 21-23 do not describe “skins”, they describe electronic tongue based in chemical sensors or cell-based sensors. The term “skins” is used repeatedly in the manuscript, but it is not conventional and should be changed to sensor or electrode.
  4. Materials and Method section lack important information. All experimental protocols should be described in details, including solution preparation, experimental conditions and equipment used.
  5. The first paragraph of the Results and Discussion is the repletion of the information already given in the Materials and methods section and can be removed.
  6. 3 c and d. How many replicated measurements were used to calculate standard deviations? Does Fig. 3e shoes the same curves as c and d?
  7. Sensor characteristics such as e.g. detection limit, lifetime should be presented.
  8. 7d. Sensor response in water should be show in the Fig. 7d.
  9. 8, line 203. What is “blended juice (red grapes and roses)”?
  10. 8, lines 204-205. Concentration of the ascorbic acid in orange and apple juice should be calculated from the sensor responses using calibration curve.
  11. Is it accidental that two juices have exactly the same ascorbic acid concentration? It would be interesting to measure some beverage that have different ascorbic acid concentration.

Round 2

Reviewer 2 Report

The manuscript has significantly improved after correction. The title now reflects the content of the paper. It can be accepted for publication after minor grammar checks and improving the introduction which, is rather too short. This can improve the scientific soundness and interest to readers.

Reviewer 3 Report

Most of the raised questions have been addressed satisfactorily and the quality of the manuscript has improved. I have only few further comments:

- The authors have corrected the sentence in the Abstract to “This work can encourage the development of wearable flexible electronics and this self-powered sour sensor has the potential that can be acted as a kind of gustatory receptors to build artificial neural networks”. Artificial neural networks are not built of receptors, ANNs are computational networks, whose structural units are called neurons, and they have nothing in common with receptors. Please correct.

- Fig. 7. It should be referred in the text that concentration of AA in the analyzed samples were calculated using equation shown in the Fig. 3. Furthermore, it would be more comprehensible if quantification results were presented not as a graph but in numeric form, i. e. as means with standard deviations.
